# The Diversity and Community Composition of Three Plants’ Rhizosphere Fungi in Kaolin Mining Areas

**DOI:** 10.3390/jof10050306

**Published:** 2024-04-24

**Authors:** Wenqi Xiao, Yunfeng Zhang, Xiaodie Chen, Ajia Sha, Zhuang Xiong, Yingyong Luo, Lianxin Peng, Liang Zou, Changsong Zhao, Qiang Li

**Affiliations:** 1Key Laboratory of Coarse Cereal Processing, Ministry of Agriculture and Rural Affairs, Sichuan Engineering & Technology Research Center of Coarse Cereal Industrialization, School of Food and Biological Engineering, Chengdu University, Chengdu 610106, China; xwq990713@126.com (W.X.); zhangyunfeng@cdu.edu.cn (Y.Z.); cxd0512@126.com (X.C.); shaajia19980108@126.com (A.S.); xiongzhuang2000@126.com (Z.X.); lyy1478963@126.com (Y.L.); penglianxin@cdu.edu.cn (L.P.); zouliang@cdu.edu.cn (L.Z.); 2School of Public Health, Chengdu Medical College, Chengdu 610500, China

**Keywords:** fungal diversity, mining activities, functional potential, kaolinite, rhizosphere fungi

## Abstract

Mining activities in the kaolin mining area have led to the disruption of the ecological health of the mining area and nearby soils, but the effects on the fungal communities in the rhizosphere soils of the plants are not clear. Three common plants (*Conyza bonariensis*, *Artemisia annua*, and *Dodonaea viscosa*) in kaolin mining areas were selected and analyzed their rhizosphere soil fungal communities using ITS sequencing. The alpha diversity indices (Chao1, Shannon, Simpson, observed-species, pielou-e) of the fungal communities decreased to different extents in different plants compared to the non-kauri mining area. The β-diversity (PCoA, NMDS) analysis showed that the rhizosphere soil fungal communities of the three plants in the kaolin mine area were significantly differentiated from those of the control plants grown in the non-kaolin mine area, and the extent of this differentiation varied among the plants. The analysis of fungal community composition showed that the dominant fungi in the rhizosphere fungi of *C. bonariensis* and *A. annua* changed, with an increase in the proportion of *Mycosphaerella* (genus) by about 20% in *C. bonariensis* and *A. annua*. An increase in the proportion of *Didymella* (genus) by 40% in *D. viscosa* was observed. At the same time, three plant rhizosphere soils were affected by kaolin mining activities with the appearance of new fungal genera *Ochrocladosporium* and *Plenodomus*. Predictive functional potential analysis of the samples revealed that a significant decrease in the potential of functions such as biosynthesis and glycolysis occurred in the rhizosphere fungal communities of kaolin-mined plants compared to non-kaolin-mined areas. The results show that heavy metals and plant species are the key factors influencing these changes, which suggests that selecting plants that can bring more abundant fungi can adapt to heavy metal contamination to restore soil ecology in the kaolin mining area.

## 1. Introduction

Kaolin is an important natural inorganic mineral that is mainly obtained from kaolinite through beneficiation and processing, and its main chemical component is silicate minerals [1,2]. This material contains high amounts of alumina and silica [3]. In nature, the formation of kaolinite is mostly related to the weathering of aluminum-containing rock minerals and the action of microorganisms. Kaolin has the characteristics of strong adsorption, good chemical stability, excellent insulation performance, and high-temperature resistance, allowing it to be widely used in industrial production [4,5]. In addition, kaolinite is widely used in construction materials, fillers, coatings, plastics, rubber, and other fields [6,7]. Kaolin can also undergo physical and chemical reactions with organic matter and other minerals to promote the formation of soil aggregates, improve soil structure, increase soil porosity, and improve soil aeration and water retention [8]. The combination of kaolinite and biochar improves the stability of biochar, reduces carbon loss, and improves soil fertility [9]. Quantitative studies of simulated soils have shown that kaolinite has a greater abundance of fungal communities than other clay minerals [10]. 

Microbes can induce the formation of clay minerals and decompose clay minerals to promote the formation of other minerals. Kaolin is a clay mineral that is produced by microorganisms and participates in their formation. Kaolin can adsorb pollutants in the soil or water environment [11,12]. The adsorption characteristics of kaolinite are also closely related to microorganisms [13]. Particulate organic matter formation and turnover are significantly influenced by fungi [14]. Kaolin mining may have multiple impacts on nearby soils, especially in relation to metal diffusion and soil quality [15]. First, there may be a risk of metal release during the mining process because kaolinite often contains metal elements such as aluminum, iron, and magnesium. The mining and extraction processes may cause these heavy metals to enter nearby soils [16]. An increase in the aluminum (Al) content in the soil may cause toxicity in the soil [17]. Mining activities may also cause physical disturbance of the soil, including soil erosion, loosening, and compaction, thus affecting the structure and permeability of the soil [18]. In addition, the discharge of wastewater, residues, and wastes generated by mining activities may also pollute the soil and groundwater [19]. These changes in soil will all affect the rhizosphere soil fungal communities of plants.

The mining activities in kaolinite mining areas will affect the physicochemical properties of nearby soil, including soil erosion, soil pollution (heavy metals adsorbed by kaolinite will be released), and land degradation [20]. These factors directly affect the ecological balance of the soil and the metabolism and activity of microbes and plants. Current studies have only focused on the biodiversity of microorganisms in soil from abandoned mining areas [21,22,23,24,25]. However, there is a lack of studies on the biodiversity of microorganisms in rhizosphere soil in mining areas. Local plants play an important role in the ecological restoration of kaolin mining areas. Plant rhizosphere fungi are important adjuvants in bioremediation and play a crucial role in ecological monitoring. Therefore, this study used ITS sequencing to reveal changes in the diversity and community composition of plant rhizosphere fungi, identify key fungal populations shared and unique to different plants, and reveal changes in the function of rhizosphere fungal communities in adapting to the environment of kaolin mining areas. From the results derived from the analysis, this study, for the first time from the perspective of plant rhizosphere fungi in kaolin mining areas, found that mining activities in kaolin mining areas reduced the diversity of local plant rhizosphere fungi, changed the species composition of rhizosphere fungi, weakened their biosynthesis and metabolism-related functions, and the rhizosphere fungi of different plants had different degrees of changes. These changes enriched our knowledge of the effects of kaolin mining on plant rhizosphere fungi and provided important references for the restoration of soil ecology in kaolin mining areas.

## 2. Methods and Materials

### 2.1. Rhizosphere Soil Sample Collection

In the kaolinite mining area near Panzhihua City, Sichuan (101.72° E, 26.58° N), the rhizosphere soil of three native plants (*Conyza bonariensis*, *Artemisia annua*, and *Dodonaea viscosa*) was sampled. Most plants have poor growth conditions in the study area, while the three plants selected in this study have a large distribution and good growth in the study area. In order to provide assistance in restoring the ecological health of the study area, we selected plants with good growth conditions to analyze their fungal diversity and species composition in the rhizosphere soil. The rhizosphere soils of *C. bonariensis*, *A. annua*, *and D. viscosa* collected from the kaolinite mining area were named Cbo, Aan, and Dvi, respectively. Three types of rhizosphere soil from non-kaolin mining areas with similar environmental conditions were used as control samples, namely, CK-Cbo, CK-Aan, and CK-Dvi. Non-rhizosphere soil taken from a non-kaolin mining area with similar environmental conditions was used as a control (KB). In order to make the samples more representative, each soil sample is a mixture of rhizosphere soil from 5 identical plants from the same region. Each sample had three replicates, totaling 21 soil samples and 105 plants. The rhizosphere soil was obtained by extracting plant roots, shaking off the soil adhering to the outside of the roots, and then packing soil into ziplock bags.

### 2.2. Genomic DNA Extraction

For fungal diversity analysis, 50 g of soil were collected for each sample. The 21 samples were then loaded into an ice pack and sent to the lab for DNA extraction and ITS sequencing. The soil samples were used to extract total genomic DNA using a soil DNA kit (D5625-02, OMEGA, Nevada County, CA, USA). After the DNA extraction, it was analyzed on a 1% (*w*/*v*) agarose gel to evaluate its quality.

### 2.3. PCR Amplification

The DNA extracted in the previous step underwent PCR amplification on the ITS1-5F region. The total DNA extracted from the rhizosphere soil samples was mixed with forward primer ITS5-1737F (5′-GGAAGTAAUGUGTCGTAACAAGG-3′), reverse primer ITS2-2043R (5′-GCTGCGTCTTCATCGATGC-3′), dNTP mixture, buffer, MgCl2, Taq DNA polymerase, and the necessary amount of nucleic acid-free pure water in preparation for PCR amplification. First, an initial denaturation step (95 °C, 3 min) was performed, and then the following process was repeated for cycle amplification: denaturation (95 °C, 30 s), annealing (50–60 °C, 30 s), and extension (72 °C, 30 s–2 min). Finally, a final extension step (5–10 min, 72 °C) was performed; then, PCR products were analyzed and purified by agarose gel electrophoresis.

### 2.4. Library Preparation, Sequencing, and Raw Data Processing

Library construction was carried out with the help of the NEBNext^®^ Ultra™ II DNA Library Prep Kit (Illumina, San Diego, CA, USA). Following the construction, the library was quantified using Qubit and Q-PCR, with the index code included in the procedure. Following that, an evaluation of the library’s quality was conducted using a Qubit@ 2.0 fluorescence instrument (Thermo Scientific, Waltham, MA, USA) and an Agilent Bioanalyzer 2100 system. Following the library’s qualification, it was sequenced using paired-end sequencing (Paired_End) on the Illumina NovaSeq platform (NovaSeq 6000), producing 200–400 bp paired-end reads. Identification of reads was carried out by matching their barcodes, followed by the removal of both the barcodes and primer sequences. The paired-end reads were then merged using FLASH V1.2.7 [26]. Readouts were performed according to QIIME V1.9.1 [27] and filtered using our quality control process. The sequences were then compared with the Silva reference database [28]. A comparison was performed to identify any chimeric sequences. After this, the sequences were excluded from the dataset. Retention of reads was contingent upon them meeting the requirements of carrying the correct barcode and forward primer sequence, having an average quality score of ≥25, and falling within the length range of 200–400 bp. The raw data were obtained after the above-described processing. The raw sequencing data contain a certain proportion of interfering data. To ensure the accuracy and reliability of the information analysis results, it is necessary to splice and filter the raw data to obtain valid data. Using the DADA2 module in QIIME2 (2019.4) [29] will perform noise reduction on the valid data and filter out the sequences with abundance of less than 5 [30] so as to obtain the final ASV [31].

### 2.5. Species Composition Analysis

The DADA2 method has been shown to outperform the traditional operational taxonomic unit (OTU) method in terms of sensitivity and specificity. This method has the capability to uncover authentic biological variation that the OTU method might have missed while also reducing the occurrence of false sequences [32]. At the same time, the replacement of OTUs with ASVs improved the precision, comprehensiveness, and repeatability of marker gene data analysis [33]. For each ASV, species annotation was conducted using the pre-trained naive Bayes classifier with the classify-sklearn algorithm of QIIME2 (2019.4) [34,35]. The species abundance table at the kingdom, phylum, class, order, family, genus, and species levels was derived from the annotation results of the ASVs and the characteristic table of each sample. Species composition and difference analysis of ASVs and cluster analysis were performed among the samples.

### 2.6. Alpha Diversity and Beta Diversity Analysis

Alpha diversity was utilized to analyze the diversity of the microbial community in the sample [36]. The richness and diversity of microbial communities within a sample can be effectively assessed through single-sample diversity analysis, which involves the use of statistical analysis indices (such as observed species, Shannon, Simpson, Chao1, Goods_coverage, dominance, and Pielou_e), species diversity curves, and dilution curves. These indices allow for the comparison of species richness and diversity among microbial communities in each sample [37,38].

Beta diversity was assessed by comparing the microbial community composition in various samples. Firstly, based on the species annotation results and the abundance information of ASVs of all samples, the information of ASVs of the same classification was combined and processed to obtain the species abundance information table (Profiling Table). At the same time, the Unifrac distance (Unweighted UniFrac) was computed based on the phylogenetic relationships among the ASVs [39,40]. The UniFrac distance is a method that uses evolutionary information between microbial sequences in each sample to calculate the intersample distance. If there are more than two samples, a distance matrix is obtained. Next, the abundance of information on ASVs was used to construct a Weighted Unifrac distance for Unifrac distance [41]. Finally, the intergroup difference analysis of the beta diversity index was conducted. Multivariate statistical methods, such as principal component analysis (PCA) and non-metric multidimensional scaling (NMDS), were used to detect differences among the various samples (groups).

### 2.7. Community Difference Analysis and Functional Prediction

Based on the obtained DNA sequence, we used the FUNGuild (http://www.funguild.org/) [42] and compared it to the MetaCyc (https://metacyc.org/) database for analysis. The results were then used to classify fungal species into different functional groups. The functional groups were classified based on the functional annotations of the fungi. Using FUNGuild, we obtained the relative abundance or expression level of each functional group in each sample. In addition, functional group annotation information was also obtained to better understand the ecological functions of the different functional groups in the fungal community. We used statistical software, such as *R* V1.2.7, to further analyze community differences in different samples.

### 2.8. Statistical Analysis

We conducted a statistical analysis to determine the degree of differences between the samples. The comparison of two groups of samples was carried out using a *t*-test, whereas samples from more than two groups were compared using a Tukey test. When the *p*-value is below 0.05, it suggests a statistically significant distinction exists among the groups.

## 3. Results

### 3.1. Quality Control and Processing of Sequencing Data

Using ITS high-throughput sequencing technology, we studied the fungi associated with *C. bonariensis*, *A. annua*, and *D. viscosa* in kaolinite mining areas and non-mining areas. As shown in Figure 1a, the sparsity curves of fungal ASVs in different samples showed that as the number of sequencing reads increased, the number of observed_species gradually increased. The curve leveled off when the number of sequencing reads approached 10,000, indicating that the number of sequencing reads was sufficient to reflect the scarcity of fungal ASVs in different samples. Figure 1b is a plot of the sparsity-based on the goods_coverage index, which can be seen to be close to 1 as the number of sequencing reads approaches 10,000. This indicates that the sequencing reads are relatively complete and have a good degree of coverage of the diversity of the fungal community in the sample, further illustrating the results mentioned above. After excluding chimeras, low-quality sequences, and short sequences, the average number of clean reads per sample for downstream analysis was 59,676.

### 3.2. Species Composition Analysis

QIIME2 (2019.4) was used to count the reads; visualize the composition distribution of each sample at the four classification levels of class, order, family, and genus; and analyze the specific species composition of each sample. As shown in Figure 2a, at the class level, we compared the 10 most abundant classes among the different samples. The fungal classes with the greatest abundance in all the samples were *Dothideomycetes*, *Sordariomycetes*, *Eurotiomycetes*, *Mortierellomycetes*, and *Tremellomycetes.* The *Dothideomycetes* in the DVI sample had the highest abundance, and the KB samples had the lowest. In general, the abundance of *Dothideomycetes* in plants grown in kaolinite mining areas was greater than that in the control plants, with a significant increase in DVI. The abundance of *Sordariomycetes* was the lowest in the DVI sample and the highest in the KB sample. In general, the abundance of *Sordariomycetes* in plants grown in kaolinite mining areas was lower than that in plants grown in non-kaolin mining areas, with significant decreases in the Cbo and DVI samples. We also found that, compared with those in plants grown in non-kaolinite mining areas, the abundance of unknown fungi in all plant samples grown in kaolinite mining areas decreased, suggesting that mining activity may affect fungal diversity in plants.

At the order level, we compared the 10 most abundant orders in different samples. Figure 2b shows that the most abundant fungal orders in all samples were *Pleosporales*, *Hypocreales*, *Capnodiales*, *Sordariales*, and *Eurotiales.* The abundance of *Pleosporales* was greatest in the DVI samples. Compared with those in plants growing in non-kaolin mining areas, the abundance of *Pleosporales* increased significantly in DVI samples. Compared to plants growing in non-kaolin mining areas, the abundance of *Hypocreales* decreased to some extent in the three plants growing in the kaolin mining area. Compared with those in plants growing in non-kaolin mining areas, the abundance of *Capnodiales* significantly increased in the Cbo and Aan samples. The abundance of *Sordariales* in the Cbo and Aan samples showed a significant decrease when compared with plants growing in non-kaolin mining areas. Similarly, at the order level, the proportions of unknown fungi among the three types of plants in the kaolinite mining area decreased.

At the family level, we compared the 10 most abundant families in different samples. As shown in Figure 2c, the most abundant fungal families in all samples were *Didymellaceae*, *Mycosphaerellaceae*, *Aspergillaceae*, *Nectriaceae*, and *Mortierellaceae*. *Didymellaceae* was the most abundant in the DVI samples. Compared to those in plants growing in non-kaolin mining areas, the abundance of *Didymellaceae* in the three plant samples from kaolinite mining areas increased, with significant increases in the Cbo and DVI samples. The abundance of *Mycosphaerellacea* significantly increased in the Cbo and Aan samples. In addition, we found that, compared with those in plants from non-kaolinite mining areas, the proportions of unknown fungal families among the three types of plants in kaolinite mining areas were lower.

At the genus level, we compared the 30 most abundant genera in different samples. As shown in Figure 2d, the fungal genera with relatively high abundance in all samples were *Didymella*, *Mycosphaerella*, *Mortierella*, *Penicillium*, *Fusarium,* and *Alternaria*. *Didymella* was very abundant in the DVI samples. Compared with those in the CK-DVI samples from non-kaolin mining areas, the abundance of *Didymella* in the DVI samples significantly increased. The abundance of *Mycosphaerella* and *Fusarium* in the Cbo samples also increased significantly, while the abundance of *Penicillium* in the Aan samples significantly decreased. Similarly, at the genus level, the proportions of unknown fungal genera among the three types of plants in the kaolinite mining area decreased.

### 3.3. Alpha Diversity

The alpha diversity indices, including observed species, Shannon, Simpson, Chao1, goods_coverage, dominance, and Pielou_e, were used to analyze the species diversity and distributional homogeneity of the rhizosphere soil fungi of the three plants. As shown in Figure 3, the observed_species, Shannon, Simpson, and Chao1 indices of the three plant samples grown in the kaolin mine were generally decreased compared to those grown in the non-kaolin mine. At the same time, the Chao1 indices of the Dvi and Aan samples significantly decreased. The Shannon index of the Aan sample decreased significantly. The Simpson’s index decreased to some extent for all three samples. The observed_species indices of the Dvi and Aan samples significantly decreased. Compared with those of the plants growing in non-kaolin mining areas, the goods_coverage indices of the three plant samples grown in kaolinite mining areas all increased, with significant increases for the Cbo and Aan samples. The pielou_e indices of the Dvi and Aan samples decreased, while that of the Cbo sample changed little.

By plotting the abundance of each ASV on the horizontal axis and the log2-transformed abundance of each ASV in each sample on the vertical axis against the abundance grade curve, we compared the abundance of each ASV in each sample, as depicted in Figure 4. The curves for the three plant samples grown in the kaolin mine were steeper and shorter on the horizontal axis than those grown in the non-kaolin mine, indicating significantly lower abundance of individual ASVs, greater variation in abundance among ASVs, and lower homogeneity.

### 3.4. Beta Diversity

We used principal coordinate analysis (PCoA) and non-metric multidimensional scaling (NMDS) based on the Bray–Curtis dissimilarity distance matrix to compare community differences between samples and used this to assess the impact of kaolin mining activities and plant species on fungal communities in plant rhizosphere soils. As shown in Figure 5a (PCoA), the fungal communities of the three plant samples yielded significant variability compared to the plant samples grown in the non-kaolinite mining area. Among the three plants from the kaolin mining area, there was less variability in the fungal communities between the Aan and Cbo samples, while Dvi and the other two plant samples had high variability in the fungal communities. This was also evident in the three plant samples grown in the non-kaolin mining area. Figure 5b: Non-metric multidimensional scaling analysis (NMDS) also shows this result, and the variability of fungal communities between Aan and Cbo samples is further reduced.

### 3.5. Community Species Composition and Difference Analysis

At the genus level, we analyzed individual samples for specific species composition and variability. Figure 6 shows the clustering heatmap. The samples were grouped together using UPGMA, with the grouping determined by the Euclidean distance of the species composition data. They were then ranked according to the clustering results, which demonstrated the differences in species composition among the samples while also reflecting the abundance of individual fungal genera present in the samples. This is similar to the results shown in Figure 2d.

To more intuitively represent the differences in species composition among the samples, we generated an ASV Venn diagram to identify the unique and shared ASVs among the samples. As shown in Figure 7, there were 44 ASVs shared between Cbo and CK-Cbo samples, 61 ASVs shared between Aan and CK-Aan samples, and 87 ASVs shared between Dvi and CK-Dvi samples. There were 37 ASVs shared between the three plant samples grown in the kaolin mine area, 300 ASVs specific to DVI, 185 to Aan, and 139 to Cbo. There were 147 ASVs shared between the three plant samples grown in the non-kaolin mine area, 610 ASVs specific to CK-DVI, 679 ASVs specific to CK-Aan, and 688 ASVs specific to CK-Cbo. There were 13 ASVs shared between all samples. We found that the number of ASVs was significantly lower in plant samples grown in the kaolin mine compared to those grown in the non-kaolin mine.

To further observe the changes in ASVs in each sample, we selected plants grown in non-kaolin mining areas as the control group and plants grown in kaolinite mining areas as the upregulation group. The fitFeature Model function was used to evaluate the distribution of each ASV via the zero-inflated log-normal model. The fitting results of the model were used to determine the significance of the differences and a Manhattan diagram was drawn. We used CK-Cbo as the control group and Cbo as the upregulated group. As shown in Figure 8a, there were significant differences for a total of 27 ASVs, of which 25 were significantly upregulated. These differences involved four phyla, nine classes, 12 orders, 13 families, and 13 genera. We used CK-Aan as the control group and Aan as the upregulation group. As shown in Figure 8b, there were significant differences for a total of five ASVs, of which three were significantly upregulated, involving one phylum, one class, one order, two families, and one genus. We used CK-Dvi as the control group and Dvi as the upregulation group. As shown in Figure 8c, a total of 18 ASVs showed significant differences, with 17 ASVs being significantly upregulated. These ASVs involved two phyla, five classes, six orders, seven families, and six genera. Most ASVs with significant differences were attributed to *Ascomycota*. Very noteworthy is the appearance of the new genera *Ochrocladosporium* and *Plenodomus* in the rhizosphere fungal communities of all three plant species growing on kaolin.

### 3.6. Community Function Prediction and Difference Analysis

Based on the MetaCyc database, we used the FUNGuild tool to annotate the fungal species of plant rhizosphere soil samples into different functional groups in order to facilitate the clarification of the effects of kaolin and plant species on the functioning of the fungal flora in plant rhizosphere soil. As shown in Figure 9, the predictions of the functional potential of the fungal flora genes in all samples were divided into five categories: biosynthesis, degradation/utilization/assimilation, generation of precursor metabolites and energy, glycan pathways, and metabolic clusters. The most abundant functional group was biosynthesis (46.22%), followed by the generation of precursor metabolites and energy (32.48%). Among the secondary functions of biosynthesis, the most abundant were nucleoside and nucleotide biosynthesis (31.19%); cofactor, prosthetic group, electron carrier, and vitamin biosynthesis (18.62%); fatty acid and lipid biosynthesis (17.26%); amino acid biosynthesis (12.07%); and carbohydrate biosynthesis (10.34%). The secondary functions of degradation/utilization/assimilation were also abundant, including fatty acid and lipid degradation (24.66%), carbohydrate degradation (23.76%), and nucleoside and nucleotide degradation (13.58%). Among the secondary functions of the generation of precursor metabolites and energy, the most abundant secondary functions were electron transfer (27.67%) and respiration (27.67%). Among the secondary functions of the metabolic clusters, the most abundant were tRNA charging (30.46%) and pyrimidine deoxyribonucleotides de novo biosynthesis I (24.16%).

Once we had the metabolic pathway abundance data in hand, we proceeded to utilize the normalized pathway abundance table to invoke the fitFeature Model function based on the grouping scenario. This function fits the distributions of the pathways using the zero-inflated log-normal model and uses the model’s fitting results to identify the significance of the differences. As a result, we analyzed the metabolic pathways with significant differences between the samples. We used the CK-Cbo sample as the control group and the Cbo sample as the upregulated group. As shown in Figure 10a, the levels of gluconeogenesis I (GLUCONEO-PWY), L-tyrosine degradation I (TYRFUMCAT-PWY), the urea cycle (PWY-4984), the superpathway of L-threonine biosynthesis (THRESYN-PWY), L-proline biosynthesis II (PWY-4981), adenosine ribonucleotide de novo biosynthesis (PWY-7219), trisphosphate biosynthesis (PWY-6351), aerobic respiration I (PWY-3781), aerobic respiration II (PWY-7279), and guanosine nucleic acid degradation II (PWY-6606) decreased significantly. We used the CK-Dvi sample as the control group and the Dvi sample as the upregulated group. As shown in Figure 10b, only the function of the superpathway of L-threonine biosynthesis (THRESYN-PWY) significantly decreased. No significant difference in functional gene expression was detected between the Aan and CK-Aan samples.

Aan was used as the control group, and Cbo was used as the upregulated group, as shown in Figure 10c, including monoacylglycerol metabolism (PWY-7420); glucose and glucose-1-phosphate degradation (GLUCOSE1PMETAB-PWY); NAD/NADP-NADH/NADPH mitochondrial interconversion NAD/NADP-NADH/NADPH cytosolic interconversion (PWY-7268); the superpathway of ubiquinol-6 biosynthesis (PWY-7235); phosphatidylglycerol biosynthesis I (PWY4FS-7); stearate biosynthesis III (PWY3O-355); sucrose degradation III (PWY-621); ubiquinol-9 biosynthesis (PWY-5871); ubiquinol-7 biosynthesis (PWY-5873); and sexual upregulation. Methyl ketone biosynthesis (PWY-7007), guanosine nucleotide degradation II (PWY-6606), the superpathway of phosphatidate biosynthesis (PWY-7411), CDP-diacylglycerol biosynthesis I (PWY-5667), aerobic respiration I (PWY-378ic1), respiration II (PWY-7279), and adenine and adenosine salvage III (PWY-6609) significantly decreased in these functional genes. Dvi was used as the control group, and Cbo was used as the upregulated group. As shown in Figure 10d, L-methionine biosynthesis III (HSERMETANA-PWY), the superpathway of pyrimidine ribonucleoside salvage (PWY-7196), and the superpathway of purine nucleic acid salvage (PWY66-409) were significantly upregulated. Octane oxidation (P221-PWY) functional genes were significantly downregulated. Analysis revealed no significant differences in functional genes between Dvi and Aan. 

When analyzing the functional potential differences among the samples of the three plants grown in the non-kaolin mine area and the KB samples, no significant differences were found. This suggests that the mining activities in the kaolin mine area have varying impacts on the functional potentials of the rhizosphere soil fungi of different plants. Therefore, new and significant variability in functional potential among the rhizosphere soil fungi of three plants grown in kaolin mining areas was generated.

## 4. Discussion

### 4.1. Rhizosphere Fungal Diversity and Community Composition

Environmental factors are key influences on fungal community composition [43]. Fungal species composition analysis of each sample, α diversity, and β diversity (observed_species, Shannon, Simpson, Chao1) results showed that multiple α indices of rhizosphere soil fungi of the plants grown in the kaolinite mining area showed a tendency to decrease and β indices of variability to widen compared to plants grown in the non-kaolinite mining area, and the levels of the fungal composition ratios changed significantly, and new genera also appeared. These changes demonstrated that kaolin mining did affect plant rhizosphere fungal diversity and community structure. It is noteworthy that the fungal species composition analyses, α-diversity, and β-diversity were not significantly different between the three plants grown in the non-kaolinite zone and the KB samples. It has been found that different fungi vary differently in different plant samples. Therefore, we conclude that mining activities and plant species in kaolin mining areas jointly affect the fungal diversity and community structure of plant rhizosphere soils [44,45]. 

At the genus level, the abundance of *Didymella*, *Mycosphaerella*, and *Fusarium* in the rhizosphere soil samples of several plants was greater. These fungi can cause plant wilt, spot disease, and fungal root rot [46,47,48]. *Didymella* was significantly more abundant in rhizosphere soil samples of Dvi plants grown on kaolin than in CK-Dvi samples, implying that kaolin may have reduced the adverse effects of this fungus on DVI. However, it could also be that DVI itself is tolerant to this type of disease and that kaolin enhances this tolerance, a characteristic exemplified by kaolin in one study [49]. *Mortierella* is involved in the decomposition and recycling of organic matter. It also produces physiologically active products such as active agents and antibiotics [50]. In one study, *Mortierella* was used in the remediation of a mine contaminated with heavy metals and was found to have a good adsorption effect on Zn [51]. The abundance of this group of *Mortierella* was significantly higher in Cbo and Aan samples than in the control group. However, the percentage of this group of fungi in Dvi samples, both kaolinite and non-kaolinite-grown, was extremely low. The emerging Ochrocladosporium (genus), which reduces the incidence of plant brown spot [52], has also been shown to inhibit pathogenic Alternaria through the production of phenolics or stem phenol derivatives [53], as evidenced by the significant decrease in the abundance of Alternaria in the rhizosphere soil of Dvi in this study. Although *Penicillium* causes some plant diseases and food spoilage [54], it also produces antibiotics such as penicillin [55] and decomposes organic matter [54]. A study shows that Penicillium species diversity in different environments and in different plants behaves differently [56]. The abundance of *Penicillium* was higher in the Cbo and Dvi samples grown in the kaolin mine than in their controls and lower in the Aan samples than in the controls, which also suggests that kaolin and plant species together influence the diversity and abundance of fungi.

### 4.2. Functions of the Rhizosphere Fungal Community

Many studies have reported that fungi have a key role in the remediation of contaminated land, for example, adsorption or immobilization of potentially toxic elements such as heavy metals [57,58], and enhancement of plant survival by increasing plant resistance and thus promoting plant growth [59,60,61,62]. We predicted changes in the functional potential of the fungal community and found that a number of functional genes involved in amino acid synthesis and degradation and sugar metabolism and the urea cycle appeared to be significantly upregulated or downregulated. Specifically, a significant decrease in L-tyrosine degradation I function occurred in the rhizosphere fungal community of Cbo samples, which implies an increase in L-tyrosine content, which has been reported in a number of studies to bind heavy metal ions by complexation and coordination [63,64,65]. Some studies have found that heavy metals interfere with sugar metabolism [66,67,68], and a significant decrease in gluconeogenesis I function occurred in the rhizosphere fungal community of Cbo samples in the present study, which demonstrates that heavy metals also affect the gluconeogenesis function of plant rhizosphere fungi. Plants accumulate proline to promote cellular osmoregulation, ROS detoxification, protection of membrane integrity, and enzyme/protein stabilization when exposed to heavy metal stress [69,70]. In our study, a significant decrease in the function of L-proline biosynthesis II (from arginine) occurred in the rhizosphere fungi of the Cbo plant, probably due to excessive proline accumulation in the plant roots. Proline synthesis in the rhizosphere fungal community was reduced to balance the proline content. L-threonine can protect cells by chelating metals [71], but the functional activity of the superpathway of L-threonine biosynthesis was reduced in the Cbo vs. Dvi samples, possibly caused by a decrease in the abundance of the fungus that dominates this function. In analyzing the functional differences between the Cbo and Aan rhizosphere fungal communities, we found that the phosphatidylglycerol biosynthesis function of the Cbo rhizosphere fungal community appeared to be significantly upregulated, and also the functional dynamics of phospholipases and phospholipid remodeling were all upregulated to some extent. The cell membrane plays a key role in the adsorption of heavy metals [72,73], and phospholipids are important components of the membrane. Heavy metals cause changes in substances related to them, such as phosphatidylglycerol, phosphatidic acid, phospholipase, and phosphatidylethanolamine [72,74], and the inter-root fungi of plants from the kaolin mining area showed significant upregulation of phosphatidylglycerol biosynthesis functions, and also the functional activities of phospholipases and phospholipid remodeling to a certain degree, which was found in each kaolin mining area sample analyzed in relation to the corresponding non-kaolin mining area samples. The inter-root fungal communities of plants in the mining area are subjected to heavy metal stress, and the cell membranes are involved in the adsorption-binding heavy metal response; therefore, the expression of their related functions rises. By analyzing the differential functions, we concluded that the key environmental factor of kaolin affecting plant inter-root fungal communities is heavy metals.

While there were no significant differences in functional potential between the three plant samples from the non-kauri site, there were significant differences in functional potential between Cbo and Aan and Cbo and Dvi from the kauri site. Based on the different analyses of metabolic pathways between plant samples grown in kaolinite and non-kaolinite mining areas, the functional genes of the Cbo rhizosphere soil fungal community were strongly affected by kaolinite. It is possible that each plant has a different fungal community in the rhizosphere soil, and different fungi are affected differently by mining activities. This result indicates that kaolinite and plant species jointly affect the functional genes of the fungal community.

Fungi have high plasticity and adaptability [75,76], and most of the fungi we observed belonged to *Ascomycota* (phylum), *Basidiomycota* (phylum), and *Mortierellomycota* (phylum). This was consistent with the results of many species composition analyses of fungi in mining areas, and the soil in the mining area provides an important shelter and nutrient supply for fungi [22,77,78,79,80]. Fungi play particularly important roles in mineral dissolution, metal and anion cycling, and the formation of free-living and symbiotic forms of minerals [81], which means that fungi can also restore the ecology of mine wastelands [82]. Some studies have shown that fungi of the genera *Mortierella*, *Penicillium*, and *Alternaria* promote plant growth under certain conditions by increasing the organic matter in the soil, improving the soil structure, supplying elements needed by plants, producing metabolites favorable to plant growth, and forming a symbiotic relationship with plants [83,84,85]. *Morterella* reduces phosphorus in the soil by desorbing phosphorus from the surface of soil mineral particles [86,87], and it also produces plant growth regulators such as IAA, gibberellic acid (GA), and ACC deaminase to improve plant stress tolerance [88,89]. *Penicillium* promotes plant growth in mining areas by producing organic acids that dissolve stable nutrients in the soil [90,91]. *Alternaria* species act by producing the same secondary metabolites as plants [92] and participate in biosynthetic pathways to promote plant growth, for example, by affecting the activities of key enzymes in the biosynthetic pathway of water-soluble phenolic acids [85,93]. Our results also indicate significant changes in functions related to biosynthesis and metabolic cycle in rhizosphere fungal communities of plants grown in mining areas, and that mining activities in kaolin mines affect the functional potential of local plant rhizosphere fungi, and that this effect varies among different plants and different fungi.

## 5. Conclusions

In our study, we found that mining activities in kaolin mines had an impact on the fungal communities of plant rhizosphere soils, generally reducing their diversity and altering fungal species composition as well as fungal community function, and the magnitude of this impact also depended on the plant species. Accordingly, in order to restore the ecological health of the soil in mining areas, plants can be purposefully selected to increase the abundance of fungi that are beneficial for environmental restoration. Our study initially depicts the effect of kaolin on the fungal community in the rhizosphere soil of plants, and the specific mechanism of the effect needs to be further investigated by considering different environments, different plants, and different fungi. Refining the relevant theoretical foundations through experimental investigations provides an important reference and significance for us to restore the ecosystem health of kaolin mining areas.

## Figures and Tables

**Figure 1 jof-10-00306-f001:**
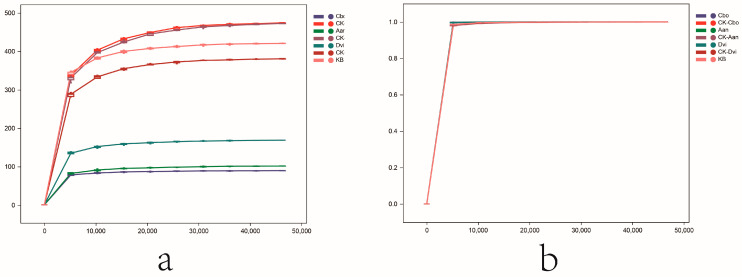
Rarefaction curve of different rhizosphere fungal samples based on observed_species (**a**) and goods_coverage (**b**).

**Figure 2 jof-10-00306-f002:**
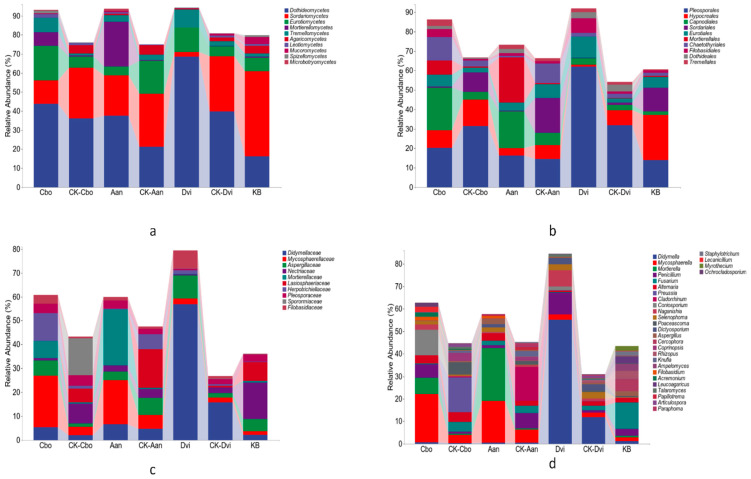
Stacked bar chart of different rhizosphere fungi species composition of classes (**a**), orders (**b**), families (**c**), and genus (**d**).

**Figure 3 jof-10-00306-f003:**
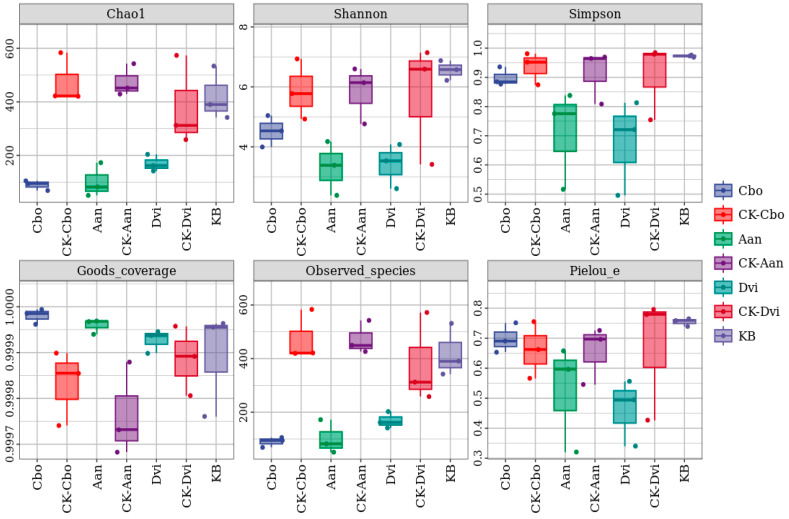
Box plots of alpha diversity indices for different rhizosphere fungal samples.

**Figure 4 jof-10-00306-f004:**
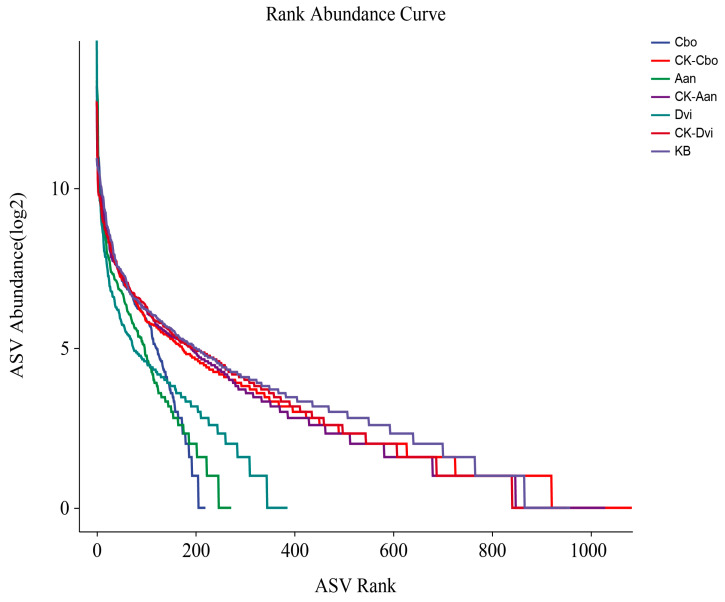
Rank abundance curves for different plant rhizosphere fungal samples by taking the abundance of ASV as the horizontal axis and the logarithmic conversion of abundance values by Log2 as the vertical axis.

**Figure 5 jof-10-00306-f005:**
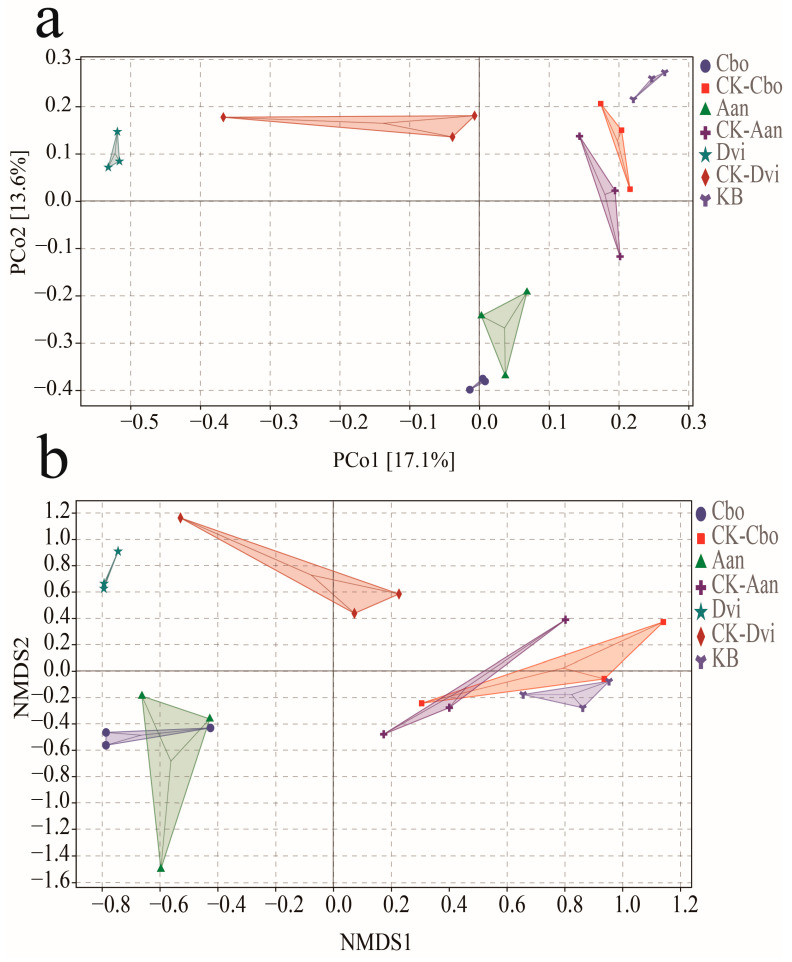
Principal coordinates analysis (**a**) and non-metric multidimensional scale analysis (**b**) of different plant rhizosphere fungal samples.

**Figure 6 jof-10-00306-f006:**
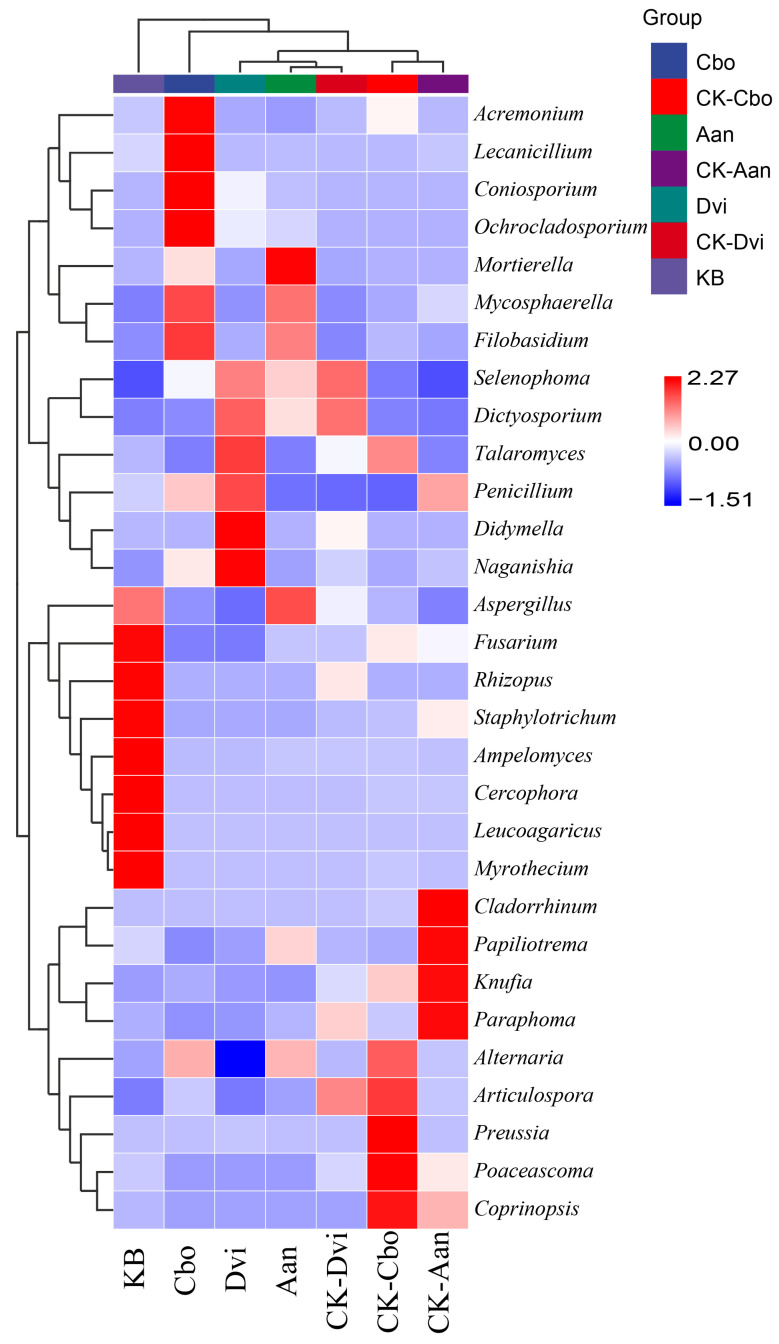
Cluster heatmap of 30 fungal genera with the highest abundance in different plant rhizosphere fungal samples.

**Figure 7 jof-10-00306-f007:**
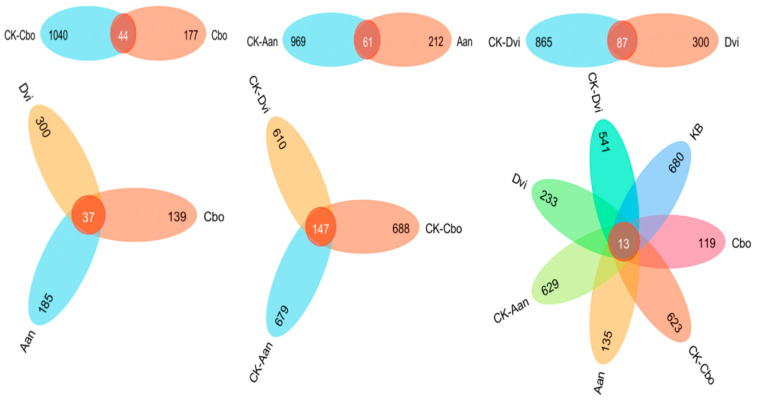
Venn diagram of shared and unique ASVs between different plant rhizosphere fungal samples.

**Figure 8 jof-10-00306-f008:**
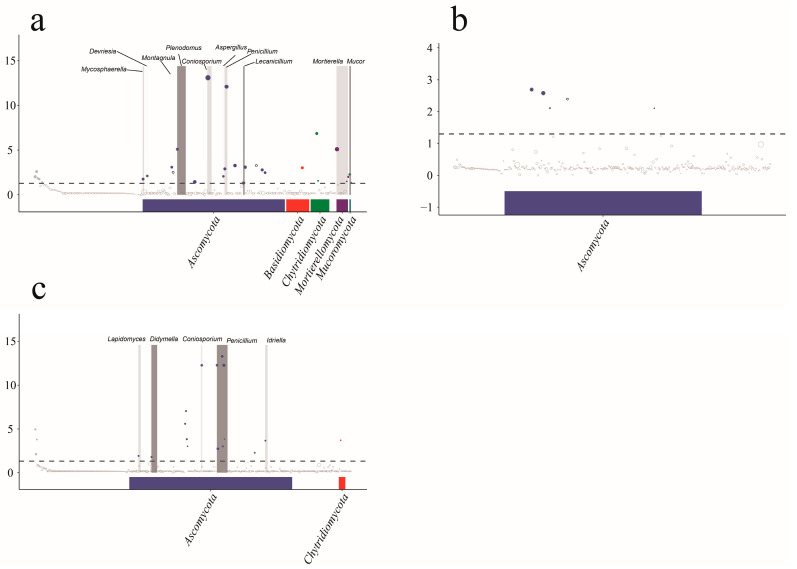
Manhattan plot of species composition differences between samples analyzed by metagenomeSeq. (**a**) Cbo vs. CK-Cbo; (**b**) Aan vs. CK-Aan; (**c**) Dvi vs. CK-Dvi. Points above the dashed line represent significant differences.

**Figure 9 jof-10-00306-f009:**
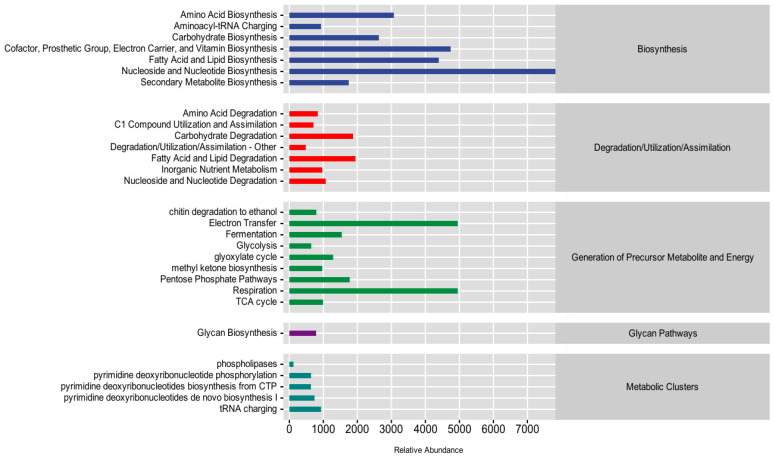
Relative abundance of metabolic pathways in plant rhizosphere fungi.

**Figure 10 jof-10-00306-f010:**
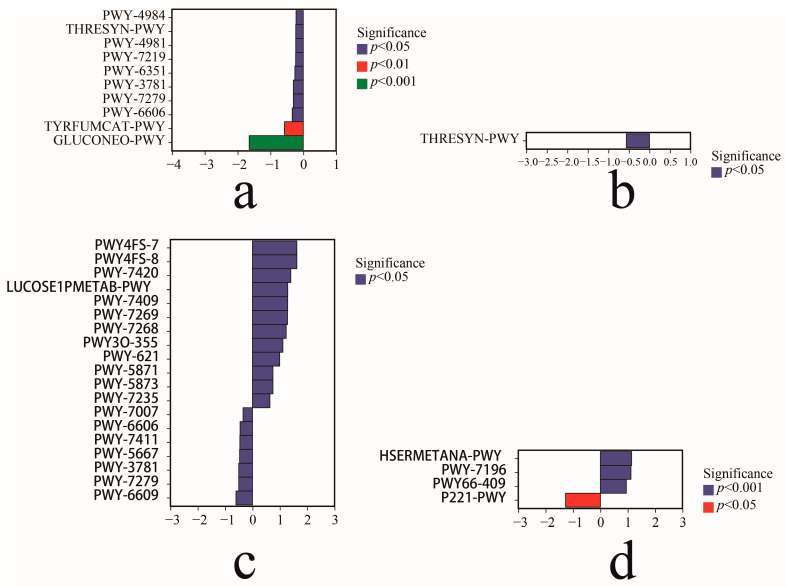
Metabolic pathways with significant Variations among different plant rhizosphere fungal samples: (**a**) Cbo vs. CK-Cbo; (**b**) Dvi vs. CK-Dvi; (**c**) Cbo vs. Aan; (**d**) Cbo vs. Dvi.

## Data Availability

Data are contained within the article.

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
