# Peer review of "The Diversity and Community Composition of Three Plants’ Rhizosphere Fungi in Kaolin Mining Areas"

_jof, 2024, doi:10.3390/jof10050306_

Round 1

Reviewer 1 Report

The article is devoted to the rather widely discussed problem in modern ecology of the influence of the mining industry on the environment, including mushrooms. This indicates the relevance of its topic. The main issues discussed in the article concern the taxonomic composition of rhizosphere fungi (at the level of classes, orders, families, and genera), as well as assessment of their abundance and ecological and physiological groups (functional features). To solve them, a modern approach to the study of soil fungi was used, based on molecular genetic methods of analysis using proven methods for their processing. Therefore, the data obtained correspond to the international level in this scientific field - soil mycology and microbiology - and the manuscript deserves publication.

However, there are several questions. 1. The work does not justify the choice of three plant species (Conyza bonariensis, Artemisia annua, Dodonaea viscosa) as objects of study. What guided the authors of the article when choosing them? 2. The title of the article talks about the effect of kaolinite mining on rhizosphere fungi. However, this is not substantiated in any way in the manuscript! In fact, the work talks about the peculiarities of the biodiversity of rhizosphere fungi in soils affected by kaolinite mining. Whether this is due to soil contamination with kaolin, or other reasons is not discussed in the work. Therefore, these articles speak only about the compositional features of rhizosphere fungi in soils susceptible to kaolin contamination. I think it would be advisable to change the title of the article! 3. Part of the introduction (lines 37-62) contains the most general information about the role of microorganisms and fungi and this part can be removed.

The manuscript may be published considering the comments made.

However, there are several questions. 1. The work does not justify the choice of three plant species (Conyza bonariensis, Artemisia annua, Dodonaea viscosa) as objects of study. What guided the authors of the article when choosing them? 2. The title of the article talks about the effect of kaolinite mining on rhizosphere fungi. However, this is not substantiated in any way in the manuscript! In fact, the work talks about the peculiarities of the biodiversity of rhizosphere fungi in soils affected by kaolinite mining. Whether this is due to soil contamination with kaolin, or other reasons is not discussed in the work. Therefore, these articles speak only about the compositional features of rhizosphere fungi in soils susceptible to kaolin contamination. I think it would be advisable to change the title of the article! 3. Part of the introduction (lines 37-62) contains the most general information about the role of microorganisms and fungi and this part can be removed.

Author Response

Dear Reviewers:

Thank you for your letter and for the reviewers’ comments concerning our manuscript entitled “Impact of kaolinite mining on the diversity of plant rhizosphere fungi” (jof-2961905). These comments are all valuable and very helpful for revising and improving our paper, as well as the important guiding significance to our researches. We have studied comments carefully and have made correction which we hope meet with approval. We have made revisions to the manuscript based on your feedback, and the revised parts are marked in red in the manuscript. we would like to show the details as follows:

  1. The work does not justify the choice of three plant species (Conyza bonariensis, Artemisia annua, Dodonaea viscosa) as objects of study. What guided the authors of the article when choosing them?

The Author's answer: We apologize for not providing a detailed explanation of the reasons for choosing plants. We have already provided an explanation. The details as follows (lines 126-130): “Most plants have poor growth conditions in the study area, while the three plants selected in this study have a large distribution and good growth in the study area. In order to provide assistance in restoring the ecological health of the study area, we selected plants with good growth conditions to analyze their fungal diversity and species composition in the rhizosphere soil.”.

  1. The title of the article talks about the effect of kaolinite mining on rhizosphere fungi. However, this is not substantiated in any way in the manuscript! In fact, the work talks about the peculiarities of the biodiversity of rhizosphere fungi in soils affected by kaolinite mining. Whether this is due to soil contamination with kaolin, or other reasons is not discussed in the work. Therefore, these articles speak only about the compositional features of rhizosphere fungi in soils susceptible to kaolin contamination. I think it would be advisable to change the title of the article!

The Author's answer: As you mentioned, this article only analyzes and discusses the diversity and Community composition of plant rhizosphere fungi growing in the soil of mining areas contaminated with kaolin. There is no evidence directly related to mining activities. Therefore, we have made revisions to the title based on your and another reviewer's feedback. The details as follows (lines 3-4): “Diversity and Community Composition of Three Plants Rhizosphere Fungi in Kaolin Mining Areas”.

  1. Part of the introduction (lines 37-62) contains the most general information about the role of microorganisms and fungi and this part can be removed.

The Author's answer: We have reviewed this paragraph again and as you mentioned, it is not directly related to this study. Another reviewer has also raised many questions about this paragraph, so we have decided to delete it according to your suggestion.

Thank you very much for taking the time to provide valuable feedback for this study. Looking forward to your reply.

Reviewer 2 Report

Dear authors,

The manuscript contains all elements of a scientific paper, however, some parts should be revised in order to improve the quality of the manuscript

Dear authors,

The abstract gives a summary of the results

In first instance: I think the title should be reformulated, example: Impact of kaolinite mining on the diversity fungi in rhizosphere different plants species.

Introduction part provides o good and comprehensive overview of the topic and background and is written straight forward. However, some parts must be revised, reformulate the sentences.

line 39 - there should be a space after the word ecosystem  and before the parentheses.

This applies to the entire manuscript.

Check if it is a good way to cite the literature.

Also, this applies to the entire manuscript.

line 40 - to rephrase this part of sentence-  Some studies have explored

line 49- the term fungal microorganisms is not well formulated

line 53 - Fungi establish themselves in plant roots, this  correct, please lines 55-61 reformulate these sentences

Аdd what is the goal of these investigations, please.

Within the Material and Methods part, the authors described the used methods. 

I would ask the authors to add details:

How many total soil samples were taken?

How many plants, what surface of soil was under the examined plant species?

The Results part is well structured and contains all results, which are clearly described in figures, but their titles are omitted,

In section Discusion shoild be the way of citing the literature should be checked.

Author Response

Dear Reviewers:

Thank you for your letter and for the reviewers’ comments concerning our manuscript entitled “Impact of kaolinite mining on the diversity of plant rhizosphere fungi” (jof-2961905). These comments are all valuable and very helpful for revising and improving our paper, as well as the important guiding significance to our researches. We have studied comments carefully and have made correction which we hope meet with approval. We have made revisions to the manuscript based on your feedback, and the revised parts are marked in red in the manuscript. we would like to show the details as follows:

1.Title section

In first instance: I think the title should be reformulated, example: Impact of kaolinite mining on the diversity fungi in rhizosphere different plants species.

The Author's answer: Another reviewer also had doubts about the title of the article. We have carefully considered the title and believe that it does not quite match the article. Therefore, we have decided to modify the title of the article. Thank you very much for providing a title that is more in line with the content of our article. We have combined the title you provided to change the title of the article. The details as follows (lines 3-4): “Diversity and Community Composition of Three Plants Rhizosphere Fungi in Kaolin Mining Areas”.

2.Introduction section

Introduction part provides o good and comprehensive overview of the topic and background and is written straight forward. However, some parts must be revised, reformulate the sentences.

2.1line 39 - there should be a space after the word “ecosystem” and before the parentheses.

This applies to the entire manuscript.

Check if it is a good way to cite the literature.

Also, this applies to the entire manuscript.

The Author's answer: Thank you very much for correcting this error for us. Your suggestions were very useful, and we have revised the entire text based on your suggestions. Once again, we express our gratitude!

2.2line 40 - to rephrase this part of sentence- Some studies have explored

2.3line 49- the term fungal microorganisms is not well formulated

2.4line 53 - Fungi establish themselves in plant roots, this correct, please lines 55-61 reformulate these sentences 

The Author's answer: Thank you very much for pointing out these three errors. Another reviewer also has many doubts about this paragraph. We have to admit that we have made many mistakes in writing this paragraph. Therefore, in order to avoid misleading readers, we have carefully considered and decided to delete this paragraph.

2.5Аdd what is the goal of these investigations, please.

The Author's answer: Thank you for your suggestion. We have provided limited elaboration on the goal of this study. Therefore, we have modified the relevant content. The details as follows (lines 108-114): “Local plants play an important role in the ecological restoration of kaolin mining areas. Plant rhizosphere fungi are important adjuvants in bioremediation and play a crucial role in ecological monitoring. Therefore, this study used ITS sequencing to reveal changes in the diversity and community composition of plant rhizosphere fungi, identify key fungal populations shared and unique to different plants, and reveal changes in the function of rhizosphere fungal communities in adapting to the environment of kaolin mining areas.”

  1. Material and Methods section

3.1How many total soil samples were taken?

3.2How many plants, what surface of soil was under the examined plant species?

The Author's answer: Thank you for your suggestion. We have supplemented the sampling content based on your suggestion. The details as follows (lines 136-141): “In order to make the samples more representative, each soil sample is a mixture of rhizosphere soil from 5 identical plants from the same region. Each sample had three replicates, totaling 21 soil samples and 105 plants. The rhizosphere soil was obtained by extracting plant roots, shaking off the soil adhering to the outside of the roots, and then packing soil into ziplock bags.”

4.Results section

The Results part is well structured and contains all results, which are clearly described in figures, but their titles are omitted.

The Author's answer: Thank you for helping us find this error. We immediately added the title of the image.

  1. Discusion section

In section Discusion shoild be the way of citing the literature should be checked.

The Author's answer: Thank you very much for your correction. We have made revisions to the literature citations in the discussion section based on your suggestions in section 2.1

Thank you very much for taking the time to provide valuable feedback for this study. Looking forward to your reply.